# DuoLink: A Dual Perspective on Link Prediction via Line Graphs

## Abstract

Link prediction remains a persistent weakness of graph neural networks (GNNs): despite strong results on node classification, decoder-based pipelines often trail simple heuristics such as Common Neighbors or Adamic–Adar. We introduce **DuoLink**, a line-graph formulation that casts link prediction as node classification on $L(G)$, where each edge-node is initialized with proximity indices and optional attribute similarity, turning heuristics into *trainable features*. This removes the encoder–decoder bottleneck, aligns message passing with edge neighborhoods, and comes with theory: we prove a 1-WL expressivity separation and an iteration-gap family showing that constant-depth models on $L(G)$ can detect edge motifs that bounded-depth decoders on $G$ cannot. Empirically, DuoLink achieves state-of-the-art performance on both homophilic and heterophilic benchmarks, with near-perfect Hits@100 on homophilic graphs and large AUC gains in heterophilic settings, consistently surpassing strong LP-GNN baselines and heuristics. By treating edges as first-class nodes, DuoLink closes—and often reverses—the long-standing gap between classical heuristics and GNNs, pointing toward unified graph models across node, edge, and graph tasks.

## 1 Introduction

Link prediction is central to numerous applications across science and industry, from friend recommendations in social networks and protein–protein interaction discovery in biology to product suggestions in recommender systems and knowledge graph completion (Liben-Nowell & Kleinberg, 2007; Guo et al., 2020). Graph Neural Networks (GNNs) are now a leading tool for learning on graphs (Kipf & Welling, 2017; Hamilton et al., 2017b), yet their performance on link prediction remains surprisingly limited. Recent benchmarks show that classical heuristics such as Common Neighbors, Adamic–Adar, Katz, and matrix factorization often outperform state-of-the-art GNN pipelines (Tola et al., 2025).

Why do GNNs that excel at node-level tasks struggle with edge-level inference? We highlight two causes. First, standard pipelines learn node embeddings and then apply a separate decoder (for example Hadamard or an MLP), creating a mismatch between representation learning and the edge prediction objective (Zhang & Chen, 2018; Wang et al., 2022b). Second, simple and interpretable heuristics capture essential edge-centric structure that GNN decoders frequently miss (Li et al., 2023).

These issues are amplified on *heterophilic graphs*, where edges often connect dissimilar nodes. While heterophily has been widely studied for node classification (Pei et al., 2020; Zhu et al., 2020), its impact on link prediction is less explored. Similarity-based decoders implicitly assume homophily, which leads to systematic errors on heterophilic links. Recent studies (Zhu et al., 2024; Di Francesco et al., 2024) emphasize that structural rather than feature-based signals are critical in such settings, yet broadly effective solutions remain elusive.

We address these gaps by reformulating link prediction as node classification on the *line graph*. Each candidate edge becomes a node with attributes derived from its local subgraph context and proximity indices. This enables end-to-end training with a single GNN or transformer on $L(G)$, aligning the model's inductive bias with the edge-level objective. Concretely, our approach (i) removes the disconnect between node embeddings and edge scoring, (ii) integrates classical heuristics as trainable inputs rather than external baselines, and (iii) applies across both homophilic and heterophilic graphs.

Beyond immediate accuracy gains, this formulation also supports the development of *graph foundational models* that share architectures and parameters across node, edge, and graph tasks without task-specific redesign (Mao et al., 2024; Liu et al., 2025). By aligning link prediction with message passing on $L(G)$, we enable practical weight sharing and consistent inductive biases across tasks.

**Our contributions**:

- We cast link prediction as node classification on the line graph, addressing core limitations of decoder-based pipelines on $G$.

- We introduce **DuoLink**, a framework where GNNs and graph transformers operate on edge representations initialized with classical proximity indices and optional attribute similarity, harmonizing parametric learning with structural signals.

- We provide theoretical support showing a 1-WL expressivity separation and an iteration-gap family where constant-depth models on $L(G)$ detect key edge motifs that bounded-depth endpoint decoders on $G$ cannot.

- Experiments on homophilic and heterophilic benchmarks show that DuoLink consistently outperforms heuristics and strong GNN baselines, often by large margins.

By treating link prediction as a native node-classification problem on line graphs and by learning over heuristic signals end to end, DuoLink narrows the gap between modern GNNs and classical methods and points toward unified, scalable, and heterophily-aware graph representation learning.

## 2 BACKGROUND

### 2.1 RELATED WORK

**GNNs and link prediction.** Graph Neural Networks (GNNs) are the standard paradigm for link prediction, where node embeddings are produced by message passing and link probabilities are estimated through simple decoders such as dot product, MLP, or distance functions (Kipf & Welling, 2016; Hamilton et al., 2017a; Guo et al., 2023). Beyond generic encoders, a rich set of link-prediction–specialized GNNs has emerged: subgraph-based models (SEAL, BUDDY (Zhang et al., 2021; Chamberlain et al., 2023)), path- and flow-based methods (Neo-GNN, NBFNet (Yun et al., 2021; Zhu et al., 2021)), count-based decoders (NCN, NCNC (Wang et al., 2023)), positional or equivariant architectures (PEG (Wang et al., 2022a)), and mixture models (Link-MoE (Ma et al., 2024)). These approaches cover a wide algorithmic spectrum, yet most continue to follow the encoder–decoder design, which can limit their ability to capture edge motifs and higher-order interactions central to link formation.

**Heterophily and link prediction.** Heterophilic graphs, where edges frequently connect dissimilar nodes, have been a focal point in node classification (Pei et al., 2020; Zhu et al., 2020; Luan et al., 2022; Platonov et al., 2023a), but remain relatively underexplored in link prediction. Classical LP methods, both heuristic and GNN-based, are grounded in homophilic priors with similarity decoders (e.g., dot product, cosine), which fail when informative links span dissimilar features (Li et al., 2023). Attention and generative models such as GAT (Veličković et al., 2018), VGAE (Kipf et al., 2016), and GIC (Mavromatis & Karypis, 2021) extend node embedding strategies, but remain constrained by similarity scoring. Recent heterophily-aware models propose alternative mechanisms: LINKX (Lim et al., 2021) decouples features from topology, DisenLink (Zhou et al., 2022) disentangles latent factors, CFLP (Zhao et al., 2022) employs counterfactual perturbations, LLP (Guo et al., 2023) propagates labels directly, and CMP (Wang et al., 2025) incorporates causal message passing. Despite these advances, general-purpose frameworks and systematic benchmarks for heterophilic link prediction remain limited.

**Line-Graph and Edge-Centric GNNs.** Several works recast link prediction on the line graph: LGNN (Cai et al., 2021) removes subgraph pooling, LGCL (Zhang et al., 2023) adds contrastive losses, and LineDi2vec (Xing & Makrehchi, 2024) extends node2vec to edges. Edge-centric GNNs instead update edges directly (e.g., EGNN (Gong & Cheng, 2019), EdgeNets (Isufi et al., 2021)).

*DuoLink is distinct in three ways.* (i) *Feature integration:* edge-nodes in $L(G)$ are initialized with classical proximity indices and attribute similarity, making heuristics trainable rather than external.

(ii) *Objective alignment:* DuoLink removes the endpoint–decoder stage, using a simple supervised classifier on $L(G)$ to align message passing with edge neighborhoods. (iii) *Theory:* we provide WL-based guarantees showing that constant-depth models on $L(G)$ capture edge motifs that bounded-depth endpoint decoders on $G$ cannot.

This combination of heuristic integration, edge-native supervision, and task-specific theory is not present in prior line-graph or edge-centric approaches, and underpins DuoLink's consistent gains on both homophilic and heterophilic benchmarks. A detailed comparison is given in Appendix B.2.

## 2.2 PAIRWISE PROXIMITY FEATURES

Before GNNs emerged, link prediction relied on proximity-based heuristics and manual feature engineering (Kumar et al., 2020; Menon & Elkan, 2011), and recent benchmarks reveal that these simple methods still rival state-of-the-art GNNs (Li et al., 2023; Tola et al., 2025). They all hinge on homophily which is the idea that structurally or semantically similar nodes are more likely to connect and fall into two categories:

**i. Local Proximity Indices.** Exploit 1– or 2–hop neighborhoods. Let $G = (V, E)$, $\mathcal{N}(u)$ the neighbors of $u$, and $k_u = |\mathcal{N}(u)|$. Common scores $S(u, v)$ include:

$$\mathcal{CN}(u,v) = |\mathcal{N}(u) \cap \mathcal{N}(v)|, \text{(Common Neighbors)} \qquad \mathcal{J}(u,v) = \frac{|\mathcal{N}(u) \cap \mathcal{N}(v)|}{|\mathcal{N}(u) \cup \mathcal{N}(v)|}, \text{(Jaccard)}$$

$$\mathcal{S}_{\cos}(u,v) = \frac{|\mathcal{N}(u) \cap \mathcal{N}(v)|}{\sqrt{k_u k_v}}, \text{(Salton/Cosine)} \qquad \mathcal{S}_{\text{S}}(u,v) = \frac{2\,|\mathcal{N}(u) \cap \mathcal{N}(v)|}{k_u + k_v}, \text{(Sørensen)}$$

$$\mathcal{AA}(u,v) = \sum_{w \in \mathcal{N}(u) \cap \mathcal{N}(v)} \frac{1}{\log k_w}, \text{(Adamic–Adar)}$$

**ii. Quasi-Local and Global Indices.** Extend local scores to 3–hop or all-walk measures, incorporate spectral information, e.g. the *Local Path* index (Lü et al., 2009; Aziz et al., 2020), *SimRank* (Jeh & Widom, 2002), and the *Katz* index (Katz, 1953).

**iii. Attribute Similarity Indices.** Beyond structural proximity, recent models add *attribute similarity* (e.g., cosine of node attributes) to form richer *pairwise features*. Fed into a lightweight classifier, these simple features still rival state-of-the-art models (Tola et al., 2025).

However, while these pairwise similarity features remain effective across many benchmarks, they (i) assume homophily, (ii) overlook higher-order structural motifs unless explicitly engineered, and (iii) lack end-to-end trainability. This motivates our line-graph model *DuoLink*, which integrates heuristic features and learned representations within a unified, differentiable framework.

## 3 DUOLINK: METHODOLOGY

In this section, we introduce *DuoLink*, a dual formulation of link prediction as node classification on the line graph, enabling effective integration of classical edge heuristics. Link prediction is traditionally posed as learning a decoder $\phi(h_u, h_v)$ over node embeddings $h_u, h_v$ obtained by message passing on $G$. This node-centric view infers edges post hoc and often misses rich edge-level structure (e.g. triangles, wedges). We instead reformulate link prediction as node classification on the line graph $L(G)$, where each original edge becomes a node and adjacent edges in $G$ become neighbors in $L(G)$ (See Figure 1). This transformation (i) enables message passing directly over edge neighborhoods, naturally capturing local motifs, and (ii) allows classical heuristics (Common Neighbors, Adamic–Adar, etc.) to be used as initial features.

In what follows, we formalize the transductive link-prediction problem and define $L(G)$, describe feature construction, present our GNN/transformer backbones, specify the prediction head and loss (Sec. 3.1), and summarize our theoretical results on expressivity and inductive bias alignment (Sec. 3.2). Throughout the paper, we focus on the common transductive link-prediction setting (Appendix B.5), predicting edges among a fixed node set, though our line-graph GNN naturally extends to inductive and semi-inductive scenarios (See Appendix B.7).

**Problem Setup and Notation.** Let $G = (V, E)$ be an undirected graph with $|V| = n$, $|E| = m$, adjacency matrix $A \in \{0,1\}^{n \times n}$, and optional node features $X \in \mathbb{R}^{n \times d}$. In the standard *transductive* link-prediction setting, a GNN learns node embeddings $H = \mathrm{GNN}(A, X)$, $H_u \in \mathbb{R}^h$, and a decoder $\phi : \mathbb{R}^h \times \mathbb{R}^h \to [0, 1]$ scores each pair $(u, v)$. This

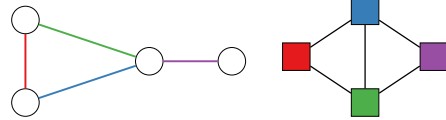

Figure 1: **Line Graphs.** Left: Graph $G$ with four edges distinguished by color. Right: Line graph $L(G)$, where each edge in $G$ becomes a node colored identically.

node-centric pipeline infers edges post hoc and may miss important edge-level structure (e.g. triangles, wedges).

To align model bias with edge reasoning, we instead build the *line graph* $L(G) = (V_L, E_L)$: $V_L = E$, $E_L = \big\{\{e, e'\} : e \neq e', \ e \cap e' \neq \emptyset\big\}$. Each original edge $e = (u, v) \in E$ becomes a node in $V_L$, and two nodes in $V_L$ are adjacent if their edges in $G$ share an endpoint (Bondy & Murty, 2008). We denote its adjacency by $A_L \in \{0,1\}^{m \times m}$ (cf. Appendix B.1).

In this reformulation, link prediction becomes node classification on $L(G)$: learn $f_L : V_L \to [0, 1]$ with $f_L(e)$ high when $e \in E$. Each $e = (u, v)$ is initialized as $z_e = \big[h_{\mathrm{struct}}(u, v) \parallel h_{\mathrm{attr}}(u, v)\big]$, combining proximity indices and (if available) attribute similarity. A GNN or GT on $(A_L, Z)$ then yields embeddings $H^{(L)}$ for classification.

**Line-Graph Transformation.** Given an undirected graph $G = (V, E)$ with $|V| = n$ and $|E| = m$, we construct its *line graph* $L(G) = (V_L, E_L)$ by treating each original edge $e = (u, v) \in E$ as a node in $V_L$. Two nodes $e = (u, v)$ and $e' = (u', v')$ in $V_L$ are connected by an edge in $E_L$ if and only if they share a common endpoint in $G$: $V_L = E$, $E_L = \big\{\{e, e'\} : e \neq e', \ e \cap e' \neq \emptyset\big\}$. Equivalently, if $B \in \{0,1\}^{n \times m}$ is the incidence matrix of $G$, then the adjacency of $L(G)$ is
$$A_L = B^{\top}B - 2I_m, \quad (A_L)_{ij} = 1 \iff e_i, e_j \text{ share an endpoint.}$$

The details are given in Appendix B.1. This transformation increases the node count from $n$ to $m$, and the edge count to $\sum_{v \in V} \binom{\deg(v)}{2}$, which is $O\big(\sum_v \deg(v)^2\big)$. In practice, for sparse graphs $(\sum_v \deg(v) = 2m)$, one can build $L(G)$ in $O(m\, d_{\max})$ time and space, where $d_{\max}$ is the maximum degree in $G$ (See Appendix B.3 for further details.).

**Geometric View of $L(G)$.** An intuitive way to see why the line-graph reformulation improves link prediction is via its simplicial-complex interpretation. Notice that for any vertex $u \in V$ of degree $k$, we induce a complete $(k-1)$-graph in $L(G)$ where its vertices correspond to edges adjacent to $u$. In particular, for each vertex $u \in V$ of degree $k$, attach a $(k-1)$-simplex $\Delta_u$ in $L(G)$ whose vertices correspond to the 1-hop neighbors of $u$ (Fig. 2). The line graph $L(G)$ is exactly the 1-skeleton of this simplicial complex $\widehat{L(G)}$: In this view, for every node $u \in V$ of degree $k$, there exists a $(k-1)$-simplex $\Delta_u \subset \widehat{L(G)}$, and the node $\widehat{e} \in V_L$ corresponding to original edge $e = (u, v)$ becomes the unique vertex in the intersection $\Delta_u \cap \Delta_v$. Furthermore, if $u$ and $v$ have common neighbors

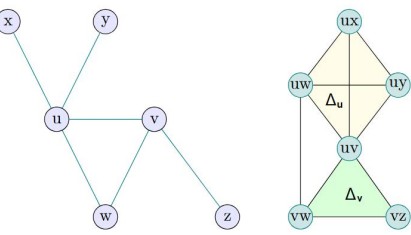

Figure 2: A graph where $u$ has degree 4 and $v$ has degree 3, sharing neighbor $w$. $L(G)$ is the 1-skeleton of the simplicial complex $\widehat{L(G)}$: a tetrahedron $\Delta_u$ on $\{uv, uw, ux, uy\}$ shaded light yellow, a triangle $\Delta_v$ on $\{uv, vw, vz\}$ shaded light green, and the extra edge between $uw$ and $vw$.

$\{w_1, \ldots, w_r\}$ in $G$, then any adjacent node $w_i \in V$ will induce an edge $\widehat{w}_i$ in $\widehat{L(G)}$ between $\Delta_u$ and $\Delta_v$, connecting the nodes $\widehat{(u, w_i)} \in \Delta_u$ and $\widehat{(v, w_i)} \in \Delta_v$ (See Figure 2). Addition of any new (or negative) edge $e' = (u', v')$ in $\mathcal{G}$ will result in the addition of a new node $\widehat{e}'$ in $V_L$ such that $\widehat{e}'$ will connect to simplices $\Delta_{u'}$ and $\Delta_{v'}$ in $\widehat{L(G)}$ becoming the unique vertex in $\widetilde{\Delta}_{u'} \cap \widetilde{\Delta}_{v'} = \widehat{e}'$.

In particular, edges with many shared neighbors or connecting substructures in $G$ induce nodes in $L(G)$ with large, richly-labeled neighborhoods. In other words, if $u$ and $v$ have $r$ common neighbors, the node $\widehat{e} = (u, v)$ in $L(G)$ links to $r$ other "edge-nodes," each representing a triangle $u$–$w_i$–$v$. This dense local neighborhood provides the GNN with direct, edge-centric evidence, rather than requiring it to infer triangles indirectly from two separate node embeddings, making it significantly easier to distinguish real links from non-links.

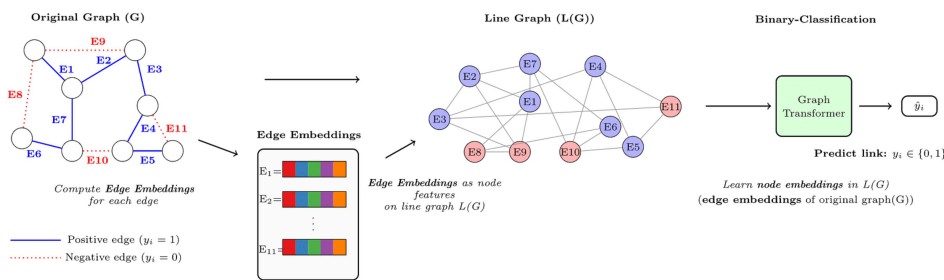

Figure 3: **DuoLink Flowchart.** Given a graph $G$ with positive (blue) and negative (red) edges, we first compute edge embeddings (proximity indices) for each potential link. These indices become node features on the line graph $L(G)$, where each original edge is treated as a node and adjacency reflects shared endpoints. A GNN or graph transformer is then applied on $L(G)$ to perform binary classification on these nodes, yielding link predictions $\hat{y}_i \in \{0, 1\}$.

By operating on $L(G)$, we enable any GNN architecture to perform message passing directly over edge neighborhoods, aligning the inductive bias with link-prediction objectives.

**Integration of Proximity Indices.** Each node $e = (u, v) \in V_L$ of the line graph is initialized with a feature vector $z_e$ that fuses *structural* and (optional) *attribute* proximity scores:
$$z_e = \big[ h_{\text{struct}}(u, v) \,\|\, h_{\text{attr}}(u, v) \big] \in \mathbb{R}^{p+q}.$$

The definitions of the proximity indices $h_{\text{struct}}(u, v)$ are provided in Sec. 2.2. The vector $h_{\text{attr}}(u, v) \in \mathbb{R}^q$ encodes similarity of node attributes (e.g. cosine or Manhattan distance on $X_u, X_v$). Further details on both the indices and the attribute similarities used in the model given in Appendix B.8.

To project $z_e$ into the GNN's hidden dimension $h$, we apply a learnable linear layer:
$$H_e^{(0)} = \phi(z_e) = \sigma\big(z_e W^{(0)} + b^{(0)}\big) \in \mathbb{R}^h,$$

where $W^{(0)} \in \mathbb{R}^{(p+q) \times h}$, $b^{(0)} \in \mathbb{R}^h$, and $\sigma$ is a nonlinearity (e.g. ReLU). These enriched features let the GNN leverage both heuristics and attributes in an end-to-end fashion.

### 3.1 GNN BACKBONE ON THE LINE GRAPH

With edge-node features $H^{(0)} \in \mathbb{R}^{m \times h}$ on $L(G)$, we apply a generic backbone $\mathcal{F}$ – either a message-passing GNN or a graph transformer – to learn refined edge embeddings:
$$H^{(\ell+1)} = \mathcal{F}^{(\ell)}\big(H^{(\ell)}, A_L\big), \quad \ell = 0, \dots, L - 1,$$

where $A_L$ is the adjacency of $L(G)$.

**Message-Passing GNNs.** A typical choice is a propagation rule of the form $H^{(\ell+1)} = \sigma\big(\tilde{D}_L^{-1/2} \tilde{A}_L H^{(\ell)} W^{(\ell)}\big)$, where $\tilde{A}_L = A_L + I$, $\tilde{D}_L = \text{diag}(\tilde{A}_L \mathbf{1})$, which covers GCN, GIN, GAT (with attention weights replacing $\tilde{A}_L$), and related variants.

**Graph Transformers.** Alternatively, one can use an attention-based transformer layer adapted to graphs:
$$H' = \text{MultiHeadAttn}\big(Q = H^{(\ell)}, K = H^{(\ell)}, V = H^{(\ell)}; A_L\big),$$
$$H^{(\ell+1)} = \text{MLP}\big(\text{LayerNorm}(H^{(\ell)} + H')\big),$$

where attention scores are masked by $A_L$ or learned from it (e.g. Graphormer, SAN).

We defer architectural specifics and hyperparameters to Sec. 4, where we instantiate $\mathcal{F}$ with several GNN and transformer models and demonstrate consistent performance gains from our line-graph reformulation.

**Prediction Head and Loss.** After $L$ layers of the backbone $\mathcal{F}$, we attach a lightweight prediction head $g : \mathbb{R}^h \to [0, 1]$, e.g. a single linear layer with sigmoid: $\quad \hat{y}_e = \sigma\big(H^L\big)$

To train, we sample a set of negative edges $\mathbb{E}^-$ uniformly from non-edges of $G$, and form the positive set $\mathbb{E}^+ = E$. The objective $\quad \mathcal{L} = -\sum_{e \in \mathbb{E}^+} \log \hat{y}_e - \sum_{e \in \mathbb{E}^-} \log\big(1 - \hat{y}_e\big) + \lambda \|W\|_2^2$, (binary cross-entropy) where $W$ collects all trainable weights and $\lambda$ is a weight-decay coefficient. At test time, we score candidate edges by $\hat{y}_e$ and rank accordingly.

## 3.2 EXPRESSIVITY AND INDUCTIVE BIAS ALIGNMENT

We formalize why operating on the line graph $L(G)$ aligns message passing with edge prediction and can separate edge patterns with strictly smaller depth than endpoint decoders on $G$.

**Setup and model classes.** Graphs are simple, finite, undirected, and connected. Unless stated, nodes have no distinguishing initial features (uniform initial color). A $t$-layer 1-WL equivalent MPNN on $G$ produces node embeddings $H = \{h_u\}_{u \in V}$. An *endpoint decoder* then scores an edge $(u, v)$ via a continuous function $\psi(h_u, h_v)$ (for example inner product, bilinear form, or an MLP on the concatenation). This hypothesis class is denoted by $\mathsf{E}_t$.

On the line graph $L(G)$, each edge $e = (u, v) \in E$ becomes a node $\hat{e}$ with initial features $z_{\hat{e}}$ computed from a constant-radius neighborhood of $(u, v)$ in $G$ (e.g., Common Neighbors, Adamic–Adar, Local Path up to $K$, attribute similarity). A $t$-layer 1-WL–equivalent MPNN on $L(G)$ consumes $z_{\hat{e}}$ and is followed by a 1-layer classifier; this class is denoted $\mathsf{L}_t$.

Two candidate edges $e_1, e_2$ are *indistinguishable* by $\mathsf{E}_t$ if every model in $\mathsf{E}_t$ assigns them the same score. Indistinguishability for $\mathsf{L}_t$ is defined analogously.

**Inductive-bias.** (Edge-neighborhood aggregation on $L(G)$) Message passing on $L(G)$ aggregates, in one round, information from edges adjacent to $e = (u, v)$, hence from the two 1-hop star neighborhoods around $u$ and $v$ viewed at the edge level. Endpoint decoders on $G$ can only combine the separately aggregated node embeddings $h_u$ and $h_v$.

**Theorem 3.1** (Expressivity separation for edge motifs)**.** *For every $t \geq 1$ there exists a graph $G_t$ and two edges $e^+, e^- \in E(G_t)$ such that*

1. *$e^+$ participates in a triangle and $e^-$ does not;*

2. *after $t$ rounds of the 1–WL color refinement on $G_t$, the endpoint colors satisfy $c_t(u) = c_t(u')$ and $c_t(v) = c_t(v')$ where $e^+ = (u, v)$ and $e^- = (u', v')$; hence every endpoint–decoder in $\mathsf{E}_t$ assigns the same score to $e^+$ and $e^-$;*

3. *there exists $t_0 \in 1, 2$ and a model in $\mathsf{L}_{t_0}$ on $L(G_t)$ that separates $\hat{e}^+$ and $\hat{e}^-$.*

Next, we prove the existence of iteration-gap family of graphs.

**Theorem 3.2** (Iteration-gap family)**.** *There exists a family $\{G_k\}_{k \geq 1}$ and edges $e_k, e'_k \in E(G_k)$ such that (i) a model in $\mathsf{L}_2$ separates $\hat{e}_k$ and $\widehat{e'_k}$ for all $k$; (ii) every model in $\mathsf{E}_k$ assigns the same score to $e_k$ and $e'_k$.*

DuoLink initializes edge-nodes with proximity indices, which can be consumed end to end on $L(G)$.

**Proposition 3.3** (Realizing proximity-index rules on $L(G)$)**.** *Let $h_{struct}(u, v) \in \mathbb{R}^p$ be any fixed collection of proximity indices computed from a bounded-radius neighborhood of $(u, v)$, for example common neighbors, Adamic–Adar, Local Path up to length $K$, or truncated Katz. Initialize each edge-node $\hat{e}$ in $L(G)$ with $z_{\hat{e}} = [h_{struct}(u, v) \| h_{attr}(u, v)]$. For any Boolean threshold rule $f$ on these indices there exists a model in $\mathsf{L}_1$ with classifier $\rho$ that realizes $f(h_{struct}(u, v), h_{attr}(u, v))$.*

**Scope and implications.** Theorems 3.1 and 3.2 do not assert that endpoint decoders can never detect motifs; rather, they show that for any fixed WL depth there exist graphs where running a shallow model on $L(G)$ separates edge patterns that bounded-depth endpoint decoders on $G$ cannot. This formalizes the benefit of our reformulation: on $L(G)$, edge motifs become first-class and are captured at constant depth.

*Link to DuoLink.* Proposition 3.3 explains why DuoLink's initialization matters: seeding edge-nodes with bounded-radius proximity indices supplies motif evidence that a 1-layer model on $L(G)$ can already realize, and further layers can refine. Together, Theorems 3.1/3.2 justify the *line-graph* component (constant-depth advantage), while Proposition 3.3 justifies the *feature-integration* component (turning heuristics into trainable signals). This theory-to-design mapping aligns with our ablations: *+ProxI* helps, but the largest gains come from the $L(G)$ reformulation that removes the endpoint–decoder bottleneck, especially on heterophilic graphs where edge-centric structure dominates.

Complete proofs and explicit constructions are in Appendix A.

## 4 EXPERIMENTS

### 4.1 EXPERIMENTAL SETUP

**Datasets.** We benchmark all methods on ten well-studied graphs that capture both homophilic and heterophilic connectivity patterns. The homophilic collection includes three widely used citation networks, CORA, CITESEER, PUBMED (Yang et al., 2016). The heterophilic collection comprises three university-webpage networks, TEXAS, WISCONSIN, CORNELL, an actor co-occurrence graph ACTOR (Pei et al., 2020; Shchur et al., 2018), and the ROMAN-EMPIRE word-dependency graph (Platonov et al., 2023b). This diverse set of datasets enables a thorough evaluation of link-prediction performance across graphs with markedly different structural and feature–label alignment properties. The details are given in Table 1.

Table 1: Homophilic and heterophilic benchmarks.

| Dataset | Nodes | Edges | Features | Classes | Edge hom. |
|---|---|---|---|---|---|
| CORA | 2,708 | 5,278 | 1,433 | 7 | 0.81 |
| CITESEER | 3,327 | 4,552 | 3,703 | 6 | 0.74 |
| PUBMED | 19,717 | 44,324 | 500 | 3 | 0.80 |
| TEXAS | 183 | 309 | 1,703 | 5 | 0.11 |
| WISCONSIN | 251 | 499 | 1,703 | 5 | 0.20 |
| CORNELL | 183 | 280 | 1,703 | 5 | 0.30 |
| ACTOR | 7,600 | 26,752 | 932 | 5 | 0.22 |
| ROMAN-EMPIRE | 22,662 | 32,927 | 300 | 18 | 0.05 |

\* Feature similarity ratio is given as there is no node classes (Zhu et al., 2024).

**Setup.** Following the experimental settings in (Kipf et al., 2016; Pan et al., 2022), we split the links in three parts: 85% training, 5% validation and 10% testing (except OGB datasets). We sample the same number of nonexisting (negative) links. For more details, see Appendix B.6.

**Hyperparameters.** For all experiments, we fixed the number of training epochs to 1000 and used a dropout rate of 0.1. For smaller datasets such as TEXAS, WISCONSIN, and CORNELL, we set the hidden dimension to 32 and used a learning rate of 0.01. Batch normalization was applied in these cases, and we tuned the number of GNN layers in the range {2, 4, 5} to assess depth sensitivity.

For the remaining (larger or more complex) datasets, we selected the hidden dimension from {64, 128}, used a learning rate of 0.001, and fixed the number of GNN layers to 5. In these settings, batch normalization was not applied.

**Computational Complexity and Runtime.** The DuoLink framework reformulates link prediction as node classification on the line graph $L(G)$, introducing only modest overhead compared to node-centric methods. Constructing $L(G)$ from an input graph $G = (V, E)$ with $|V| = n$ and $|E| = m$ takes $O(m, d_{\max})$ time and space, where $d_{\max} = \max_{v \in V} \deg(v)$, efficient for sparse graphs. In $L(G)$, the number of nodes is $m$ and the number of edges is $\sum_{v \in V} \binom{\deg(v)}{2}$, bounded by $O(m, d_{\max})$. Each GNN layer on $L(G)$ thus incurs $O(|E_L|)$ cost, matching per-layer complexity of a standard GNN on $G$ when $|E_L| = O(m, d_{\max})$. Computing classical edge heuristics (e.g., Common Neighbors, Adamic–Adar) for initialization likewise requires $O(m, d_{\max})$ time. Hence, for sparse real-world networks where $m \gg n$ but $d_{\max}$ stays small, DuoLink preserves linear scaling in the size of the original graph while enabling richer edge-centric learning.

We ran encoding extraction experiments on a single machine with 12th Generation Intel Core i7-1270P vPro Processor (E-cores up to 3.50 GHz, P-cores up to 4.80 GHz), and 32GB of RAM (LPDDR5-6400MHz). It took 2 minutes and 40 seconds to retrieve the encodings for PUBMED dataset. The classification experiment were conducted on a virtual HPC node equipped with an NVIDIA H100 NVL GPU (94 GB, PCIe), two AMD EPYC 9334 CPUs (2.7 GHz, 32 cores each), and 768 GB of RAM. Under this configuration, vanilla SAGE completed training in 3 minutes and 52 seconds, while DL-SAGE required 13 minutes and 31 seconds for PUBMED dataset. The code is available at `https://anonymous.4open.science/r/DuoLink-061D`.

### 4.2 RESULTS

**DuoLink Improvements.** Table 2 shows DuoLink delivers consistent, large gains over vanilla GNN and transformer backbones (GCN (Kipf & Welling, 2016),GIN (Xu et al., 2019),GSAGE (Hamilton et al., 2017a), DeepGCN (Li et al., 2020),GatedGraph (Li et al., 2016),SGFormer (Ren et al., 2023) and Polynormer (Deng et al., 2024)) in both homophilic (Hits@100) and heterophilic (AUC) settings. Standalone performance of heuristic features (ProxI) is already strong, but DuoLink not only recovers those signals but exceeds them, often with **double-digit average improvements**, especially on heterophilic benchmarks. A key reason is the conversion of link prediction into node classification

Table 2: **DuoLink Improvements.** Comparison of vanilla models, their proximity-index fused variants (ProxI), and line-graph reformulation (DuoLink) on homophilic (Hits@100) and heterophilic (AUC) benchmarks. Rightmost columns show average gains of ProxI and DuoLink over vanilla.

| Method | CORA 0.81 Hits@100 | CITESEER 0.74 Hits@100 | PUBMED 0.80 Hits@100 | Ave. Imp. | ACTOR 0.22 AUC | CORNELL 0.30 AUC | TEXAS 0.11 AUC | WISCONSIN 0.20 AUC | ROMAN 0.05 AUC | Ave. Imp. |
|---|---|---|---|---|---|---|---|---|---|---|
| ProxI+MLP | $93.11_{\pm0.77}$ | $95.39_{\pm0.37}$ | $75.53_{\pm0.42}$ | – | $87.38_{\pm0.23}$ | $77.40_{\pm7.55}$ | $85.26_{\pm1.93}$ | $82.28_{\pm2.94}$ | $88.92_{\pm1.11}$ | – |
| GCN | $66.27_{\pm1.49}$ | $58.53_{\pm3.03}$ | $51.08_{\pm0.84}$ | – | $84.78_{\pm0.27}$ | $73.50_{\pm3.84}$ | $78.48_{\pm4.17}$ | $79.44_{\pm2.42}$ | $81.28_{\pm0.57}$ | – |
| GCN+ProxI | $84.54_{\pm1.55}$ | $87.81_{\pm1.17}$ | $73.34_{\pm1.78}$ | 23.27 | $88.04_{\pm0.50}$ | $80.05_{\pm3.90}$ | $86.57_{\pm1.83}$ | $86.57_{\pm1.83}$ | $86.21_{\pm2.01}$ | 9.98 |
| DL-GCN | $80.64_{\pm2.85}$ | $72.74_{\pm2.37}$ | $51.69_{\pm3.19}$ | 9.73 | $89.19_{\pm0.79}$ | $72.24_{\pm12.04}$ | $80.51_{\pm3.52}$ | $86.77_{\pm4.71}$ | $85.87_{\pm5.49}$ | 5.70 |
| GIN | $49.20_{\pm4.93}$ | $41.73_{\pm2.29}$ | $39.14_{\pm1.80}$ | – | $82.10_{\pm0.67}$ | $73.37_{\pm4.66}$ | $79.05_{\pm5.27}$ | $76.48_{\pm2.80}$ | $74.40_{\pm0.81}$ | – |
| GIN+ProxI | $85.13_{\pm1.11}$ | $86.1_{\pm2.20}$ | $73.82_{\pm4.23}$ | 38.33 | $87.67_{\pm0.72}$ | $81.13_{\pm3.28}$ | $86.44_{\pm2.33}$ | $86.17_{\pm2.14}$ | $90.83_{\pm1.34}$ | 15.61 |
| DL-GIN | $79.67_{\pm3.85}$ | $75.16_{\pm3.63}$ | $57.53_{\pm1.82}$ | 27.43 | $90.98_{\pm0.35}$ | $75.13_{\pm5.20}$ | $85.72_{\pm3.86}$ | $83.72_{\pm2.97}$ | $94.03_{\pm1.16}$ | 14.73 |
| SAGE | $60.06_{\pm3.24}$ | $60.00_{\pm4.07}$ | $56.25_{\pm2.23}$ | – | $83.34_{\pm0.66}$ | $69.66_{\pm6.03}$ | $73.03_{\pm5.19}$ | $69.69_{\pm2.78}$ | $80.30_{\pm0.95}$ | – |
| SAGE+ProxI | $71.40_{\pm3.41}$ | $59.00_{\pm4.49}$ | $65.60_{\pm2.69}$ | 6.56 | $87.55_{\pm0.43}$ | $72.70_{\pm6.00}$ | $76.91_{\pm4.77}$ | $79.60_{\pm1.79}$ | $89.99_{\pm1.48}$ | 10.24 |
| DL-SAGE | $98.11_{\pm0.73}$ | $97.09_{\pm0.70}$ | $96.68_{\pm0.53}$ | 38.52 | $98.52_{\pm0.23}$ | $79.09_{\pm1.63}$ | $85.80_{\pm4.97}$ | $88.74_{\pm2.17}$ | $98.36_{\pm0.45}$ | 24.83 |
| DeepGCN | $49.18_{\pm6.03}$ | $54.54_{\pm6.13}$ | $53.10_{\pm2.68}$ | – | $83.23_{\pm0.78}$ | $70.00_{\pm4.50}$ | $74.25_{\pm4.99}$ | $71.48_{\pm2.71}$ | $80.13_{\pm0.56}$ | – |
| DeepGCN+ProxI | $72.58_{\pm3.16}$ | $75.99_{\pm2.87}$ | $69.43_{\pm1.19}$ | 20.39 | $87.72_{\pm0.38}$ | $75.83_{\pm4.63}$ | $76.64_{\pm5.42}$ | $79.86_{\pm2.45}$ | $90.59_{\pm0.70}$ | 10.52 |
| DL-DeepGCN | $96.81_{\pm0.37}$ | $92.46_{\pm1.30}$ | $92.04_{\pm1.26}$ | 41.49 | $97.73_{\pm0.24}$ | $79.19_{\pm5.58}$ | $88.14_{\pm2.67}$ | $90.74_{\pm2.18}$ | $97.09_{\pm0.75}$ | 24.60 |
| GatedGraph | $27.08_{\pm3.19}$ | $33.03_{\pm4.61}$ | $45.70_{\pm3.75}$ | – | $82.49_{\pm1.21}$ | $69.68_{\pm4.72}$ | $71.91_{\pm5.32}$ | $71.87_{\pm4.11}$ | $76.89_{\pm2.66}$ | – |
| GatedGraph+ProxI | $77.40_{\pm2.30}$ | $66.36_{\pm7.07}$ | $72.91_{\pm2.02}$ | 36.95 | $87.57_{\pm0.97}$ | $74.30_{\pm5.14}$ | $78.05_{\pm4.78}$ | $80.75_{\pm2.73}$ | $89.65_{\pm1.48}$ | 12.49 |
| DL-GatedGraph | $97.25_{\pm0.90}$ | $95.90_{\pm1.94}$ | $92.21_{\pm2.40}$ | 59.85 | $97.74_{\pm0.61}$ | $76.38_{\pm5.54}$ | $85.18_{\pm5.35}$ | $88.44_{\pm3.09}$ | $98.52_{\pm0.38}$ | 24.47 |
| SGFormer | $39.66_{\pm7.85}$ | $44.12_{\pm7.13}$ | $51.86_{\pm3.55}$ | – | $82.90_{\pm1.17}$ | $67.63_{\pm5.01}$ | $74.10_{\pm4.77}$ | $71.22_{\pm3.08}$ | $76.23_{\pm1.66}$ | – |
| SGFormer+ProxI | $73.08_{\pm3.36}$ | $69.79_{\pm2.49}$ | $67.99_{\pm3.07}$ | 25.07 | $86.92_{\pm0.93}$ | $72.35_{\pm6.33}$ | $77.35_{\pm5.05}$ | $80.20_{\pm1.77}$ | $89.54_{\pm0.91}$ | 11.43 |
| DL-SGFormer | $97.61_{\pm0.55}$ | $94.16_{\pm1.84}$ | $94.04_{\pm1.23}$ | 50.06 | $97.52_{\pm0.77}$ | $74.07_{\pm2.54}$ | $84.90_{\pm3.29}$ | $86.58_{\pm2.29}$ | $97.51_{\pm0.62}$ | 22.83 |
| Polynormer | $42.58_{\pm6.40}$ | $51.93_{\pm6.96}$ | $59.94_{\pm2.84}$ | – | $81.73_{\pm0.19}$ | $70.38_{\pm4.76}$ | $73.45_{\pm4.66}$ | $72.25_{\pm3.51}$ | $80.89_{\pm0.73}$ | – |
| Polynormer+ProxI | $75.44_{\pm3.04}$ | $84.87_{\pm4.02}$ | $67.06_{\pm1.81}$ | 24.31 | $87.41_{\pm0.31}$ | $71.85_{\pm5.64}$ | $80.76_{\pm5.27}$ | $82.05_{\pm1.37}$ | $89.74_{\pm0.51}$ | 11.04 |
| DL-Polynormer | $98.42_{\pm0.51}$ | $97.70_{\pm0.87}$ | $94.29_{\pm0.69}$ | 45.32 | $98.60_{\pm0.22}$ | $81.19_{\pm7.34}$ | $85.67_{\pm6.55}$ | $88.48_{\pm3.11}$ | $99.49_{\pm0.16}$ | 24.91 |

on the line graph, where GNNs are naturally powerful; this reframing lets message passing operate directly over edge neighborhoods and exposes structural motifs that decoder-based pipelines miss.

DuoLink's strength lies in embedding classical heuristics as trainable edge-node features and refining them through end-to-end learning. By reframing link prediction as node classification, it aligns the model's inductive bias with edge-level inference, combining the interpretability of ProxI with the flexibility of deep learning. This fusion is especially effective in heterophilic settings, where structural cues outweigh similarity.

**DuoLink vs. SOTA on Homophilic Benchmarks.** For the **homophilic setting**, we compare DuoLInk with three families of approaches: *embedding methods* (Node2vec (Grover & Leskovec, 2016), Matrix Factorization (Menon & Elkan, 2011), MLP), *standard GNNs* (GCN (Kipf & Welling, 2016), GAT (Veličković et al., 2018), GSAGE (Hamilton et al., 2017a), GAE (Kipf et al., 2016)), and *specialized link-prediction GNNs* (SEAL (Zhang et al., 2021), Neo-GNN (Yun et al., 2021), NBFNet (Zhu et al., 2021), PEG (Wang et al., 2022a), BUDDY (Chamberlain et al., 2023), NCN/NCNC (Wang et al., 2023), Link-MoE (Ma et al., 2024)).

Table 3 shows that DuoLink establishes a new state of the art on homophilic link prediction benchmarks. Both DL-SAGE and DL-Polynormer achieve *near-perfect* Hits@100 on Cora, Citeseer, and Pubmed, clearly outperforming all prior methods, including the most competitive LP-GNNs such as SEAL, BUDDY, Neo-GNN, NCN/NCNC, NBFNet, PEG, and Link-MoE. As highlighted in Table 2, these gains stem from the line-graph reformulation itself, *rather than from heuristic augmentation*, since it elevates edge motifs to first-class learning targets and incorporates proximity indices as trainable signals. This design removes the endpoint–decoder bottleneck of conventional pipelines and equips the model with direct structural evidence for true links. Even against the strongest recent baselines, DuoLink consistently delivers higher accuracy with lower variance, underscoring its robustness. These

Table 3: **Homophilic benchmarks.** Link prediction results (Hits@100). The top three models are highlighted: First, Second, and Third.

| Models | CORA | CITESEER | PUBMED |
|---|---|---|---|
| **Node2Vec** | $84.88_{\pm0.96}$ | $89.89_{\pm1.48}$ | $63.07_{\pm0.34}$ |
| **MF** | $66.39_{\pm5.03}$ | $59.47_{\pm2.69}$ | $53.75_{\pm2.06}$ |
| **MLP** | $85.52_{\pm1.44}$ | $91.25_{\pm1.90}$ | $84.19_{\pm1.33}$ |
| **GCN** | $91.29_{\pm1.25}$ | $91.74_{\pm1.24}$ | $87.41_{\pm0.65}$ |
| **GAT** | $90.70_{\pm1.03}$ | $91.69_{\pm1.21}$ | $80.95_{\pm0.72}$ |
| **SAGE** | $91.00_{\pm1.52}$ | $96.50_{\pm0.53}$ | $90.02_{\pm0.70}$ |
| **GAE** | $92.75_{\pm0.95}$ | $95.17_{\pm0.50}$ | $84.30_{\pm0.31}$ |
| **SEAL** | $84.76_{\pm1.16}$ | $85.60_{\pm2.71}$ | $76.06_{\pm4.12}$ |
| **BUDDY** | $91.42_{\pm1.26}$ | $95.40_{\pm0.63}$ | $83.21_{\pm0.59}$ |
| **Neo-GNN** | $87.76_{\pm1.37}$ | $89.10_{\pm0.97}$ | $86.12_{\pm1.18}$ |
| **NBFNet** | $88.63_{\pm0.46}$ | $86.68_{\pm0.42}$ | $79.18_{\pm0.71}$ |
| **PEG** | $91.42_{\pm0.80}$ | $94.82_{\pm0.81}$ | $76.45_{\pm3.83}$ |
| **NCN** | $95.56_{\pm0.79}$ | $96.17_{\pm1.06}$ | $90.43_{\pm0.64}$ |
| **NCNC** | $95.62_{\pm0.84}$ | $97.54_{\pm0.59}$ | $91.93_{\pm0.60}$ |
| **Link-MoE** | $96.26_{\pm0.09}$ | $96.44_{\pm0.14}$ | $90.38_{\pm0.24}$ |
| **DL-SAGE** | $98.11_{\pm0.73}$ | $97.09_{\pm0.70}$ | $96.68_{\pm0.53}$ |
| **DL-Polynormer** | $98.42_{\pm0.51}$ | $97.70_{\pm0.87}$ | $94.29_{\pm0.69}$ |

findings confirm that aligning message passing with edge neighborhoods not only bridges the long-standing gap with heuristics but also secures a decisive advantage for homophilic link prediction.

**DuoLink vs. SOTA on Heterophilic Benchmarks.** In the **heterophilic setting**, we compare DuoLink against *attention and generative* baselines (GAT (Veličković et al., 2018), VGAE (Kipf et al., 2016), GIC (Mavromatis & Karypis, 2021)) and *heterophily-aware* models (LINKX (Lim et al., 2021), DisenLink (Zhou et al., 2022), CFLP (Zhao et al., 2022), LLP (Guo et al., 2023), CMP (Wang et al., 2025)).

On heterophilic benchmarks, DuoLink variants (DL-SAGE, DL-Polynormer) far outperform both standard baselines (GAT, VGAE, GIC) and specialized heterophily-aware methods (LINKX, DisenLink, CFLP, LLP), often by very large margins in AUC. This gap highlights that conventional similarity or feature-decoupling assumptions break down in heterophilic settings, whereas DuoLink's line-graph reformulation gives GNNs direct access to edge-centric structural context,

Table 4: **Heterophilic Benchmarks.** Link prediction results (AUC%). The top three models are highlighted: First, Second, and Third.

| Model | ACTOR | CORNELL | TEXAS | WISCONSIN | ROMAN |
|---|---|---|---|---|---|
| GAT | $67.80_{\pm1.12}$ | $61.13_{\pm3.23}$ | $65.73_{\pm5.06}$ | $68.10_{\pm4.40}$ | $83.34_{\pm0.34}$ |
| VGAE | $70.82_{\pm0.81}$ | $58.18_{\pm9.47}$ | $66.75_{\pm10.09}$ | $71.30_{\pm4.60}$ | $73.27_{\pm0.83}$ |
| GIC | $70.29_{\pm0.29}$ | $58.01_{\pm3.41}$ | $66.19_{\pm7.32}$ | $75.24_{\pm8.45}$ | $56.80_{\pm3.83}$ |
| LINKX | $72.13_{\pm1.04}$ | $59.43_{\pm4.17}$ | $71.92_{\pm3.82}$ | $80.10_{\pm3.80}$ | $69.23_{\pm0.95}$ |
| DisenLink | $59.19_{\pm0.48}$ | $60.71_{\pm5.10}$ | $77.88_{\pm4.03}$ | $84.40_{\pm1.90}$ | $67.66_{\pm0.84}$ |
| CFLP | $80.41_{\pm0.32}$ | $73.14_{\pm5.42}$ | $66.02_{\pm3.84}$ | $79.14_{\pm4.89}$ | OOM |
| LLP | $80.37_{\pm1.07}$ | $68.20_{\pm7.96}$ | $71.88_{\pm3.95}$ | $67.43_{\pm0.40}$ | $82.63_{\pm3.48}$ |
| CMP | $86.81_{\pm0.55}$ | $73.59_{\pm5.38}$ | $79.26_{\pm5.38}$ | NA | NA |
| DL-SAGE | $98.52_{\pm0.23}$ | $79.09_{\pm1.63}$ | $85.80_{\pm4.97}$ | $88.74_{\pm2.17}$ | $98.36_{\pm0.45}$ |
| DL-Polynormer | $98.60_{\pm0.22}$ | $81.19_{\pm7.34}$ | $85.67_{\pm6.55}$ | $88.48_{\pm3.11}$ | $99.49_{\pm0.16}$ |

\* NA means code is not available, OOM means Out of Memory.

letting proximity heuristics be refined through message passing over richly connected neighborhoods. By casting link prediction as node classification on the line graph and initializing edge nodes with classical proximity indices, DuoLink aligns inductive bias with the true inference granularity. That alignment is especially powerful in heterophilic settings, where edge formation depends on higher-order connectivity patterns rather than feature similarity, making the fusion of heuristics and learned representations critical for reliable prediction.

**t-SNE Visualizations.** We provide t-SNE visualizations in Appendix B.9 to compare test-set edge representations from raw heuristics, standard GNNs, and DuoLink. As shown in Figures 4a to 4d, DuoLink achieves much clearer separation of positive and negative edges, demonstrating the benefit of edge-centric learning on the line graph.

**Limitations.** While our approach shows strong performance and broad applicability, it introduces some additional preprocessing due to line graph construction, which may impact scalability on very large or dense graphs. However, this overhead is manageable in all evaluated settings, and efficient construction strategies can further mitigate it. Our current experiments focus on static graphs, but the framework is general and can be extended to dynamic or temporal graphs in future work. Although we incorporate classical heuristics to enhance performance, the model remains flexible and effective even when such features are absent or limited.

## 5 CONCLUSION

We introduced *DuoLink*, a line-graph formulation that casts link prediction as node classification on $L(G)$ and treats classical proximity indices and attribute similarity as *trainable edge-node inputs*. This closes the encoder–decoder gap, comes with WL-based guarantees (expressivity separation and an iteration-gap family), and yields consistent gains on homophilic and heterophilic benchmarks. Beyond showing improvements beyond heuristics alone, DuoLink clearly departs from prior line-graph methods by integrating heuristics inside the model and aligning message passing with edge neighborhoods under a task-specific theory. Looking ahead, we aim to extend DuoLink to *dynamic/temporal* and *heterogeneous* graphs (via typed line graphs), integrate it into *graph foundational models* with self-supervised objectives on $L(G)$, and tighten motif-level theory while scaling with sparse, memory-aware implementations for large real-world graphs.

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

# Appendix

## A  PROOFS OF THEOREMS

In this part, we provide complete proofs of the theoretical results stated in Section 3.2. The main goal is to rigorously justify the separation claims between endpoint-based link predictors and line-graph-based models. We first establish the expressivity separation for edge motifs (Theorem 3.1), which shows that shallow GNNs on $L(G)$ can capture local edge patterns invisible to bounded-depth endpoint decoders on $G$. We then extend this to an iteration-gap family (Theorem 3.2), where line-graph models with constant depth succeed while endpoint decoders require depth that grows with the graph construction. Finally, we prove Proposition 3.3, which formalizes how common proximity indices can be encoded as initial features on $L(G)$ and realized by a single WL layer and linear classifier. Together these results provide the mathematical underpinnings for our claim that DuoLink aligns inductive bias with edge-centric learning and supports practical feature integration.

**Theorem 3.1.** *(Expressivity separation for edge motifs) For every $t \geq 1$ there exists a graph $G_t$ and two edges $e^+, e^- \in E(G_t)$ such that*

1. *$e^+$ participates in a triangle and $e^-$ does not;*
2. *after $t$ rounds of the 1–WL color refinement on $G_t$, the endpoint colors satisfy $c_t(u) = c_t(u')$ and $c_t(v) = c_t(v')$ where $e^+ = (u, v)$ and $e^- = (u', v')$; hence every endpoint–decoder in $\mathsf{E}_t$ assigns the same score to $e^+$ and $e^-$;*
3. *there exists $t_0 \in 1, 2$ and a model in $\mathsf{L}_{t_0}$ on $L(G_t)$ that separates $\hat{e}^+$ and $\hat{e}^-$.*

*Proof of Theorem 3.1.*     To establish the separation, we explicitly construct for each depth $t$ a graph $G_t$ containing two marked edges: one in a triangle and one not. We then use regular tree padding to guarantee that, after $t$ rounds of 1–WL, the endpoint colors of both edges remain indistinguishable, so any endpoint decoder scores them equally. Finally, we show that in the line graph $L(G_t)$ the local neighborhoods of the two edge-nodes differ by a simple cross-edge pattern, allowing a shallow WL refinement on $L(G_t)$ to separate them.

**Construction of $G_t$.** Fix $t \geq 1$ and an integer $\Delta \geq 3$.

*Triangle core.* Create vertices $u, v, w$ and edges $(u, v), (v, w), (w, u)$. Attach $(\Delta - 2)$ disjoint leaf roots to each of $u, v, w$ and then replace each such leaf by the root of an identical $(\Delta - 1)$-ary tree of depth $t - 1$ (every internal node has total degree $\Delta$). Denote the marked edge by $e^+ = (u, v)$.

*Diamond core.* Create vertices $u', v', a, c$ and edges $(u', v'), (u', a), (a, c), (c, v')$ so that $u', a, c, v'$ forms a 4-cycle with diagonal $(u', v')$. Attach $(\Delta - 2)$ disjoint leaf roots to each of $u', v', a, c$ and replace each leaf by the root of an identical $(\Delta - 1)$-ary tree of depth $t - 1$ as above. Denote the marked edge by $e^- = (u', v')$.

This yields the graph $G_t$ which contains the two marked edges $e^+$ and $e^-$.

**Step 1.**     By construction, $e^+$ participates in the triangle $(u, v, w)$, while $e^-$ has no common neighbor and therefore is in no triangle. We use the following standard fact about color refinement on regular tree paddings.

**Lemma A.1** (Synchronized padding)**.** *Let $T_{\Delta, D}$ denote the rooted $(\Delta - 1)$-ary tree of depth $D$ with uniform initial color on all vertices and total degree $\Delta$ for every internal node. For any two copies of $T_{\Delta, D}$, for all rounds $s = 0, 1, \ldots, D$ all nodes at the same depth have the same 1–WL color at round $s$. Moreover, if two host vertices $x$ and $y$ have the same round-$s$ color and each is attached to the roots of $m$ disjoint copies of $T_{\Delta, D}$, then after one more WL round the multisets of neighbor colors at $x$ and $y$ that come from the attached trees are identical.*

*Proof.* The claim for $T_{\Delta, D}$ follows by induction on $s$. At round 0 all colors are equal. If nodes at depth $d$ have equal colors at round $s$, then a node at depth $d - 1$ sees the same multiset of round-$s$ colors from its children and the same number of neighbors overall, hence nodes at depth $d - 1$ get equal colors at round $s + 1$. The second statement follows because attached trees evolve independently and symmetrically and contribute identical neighbor color multisets to $x$ and $y$ when $x$ and $y$ share the same round-$s$ color. $\square$

**Step 2.** Run 1–WL color refinement on $G_t$ with uniform initial colors. We show that after $t$ rounds,

$$c_t(u) = c_t(u') \quad \text{and} \quad c_t(v) = c_t(v').$$

We establish a stronger invariant by induction on $s = 0, 1, \ldots, t$:

$$\mathcal{C}_1^{(s)} = \{u, u'\}, \qquad \mathcal{C}_2^{(s)} = \{v, v'\}, \qquad \mathcal{C}_3^{(s)} = \{w, a, c\},$$

such that every vertex in $\mathcal{C}_i^{(s)}$ has the same round-$s$ color, and every node inside an attached tree has a round-$s$ color that depends only on its depth and on which class $\mathcal{C}_i^{(s)}$ its root is attached to.

The base case $s = 0$ is trivial. Assume the claim holds at round $s$. Consider $u$ and $u'$. Each has one neighbor in $\mathcal{C}_2^{(s)}$, one neighbor in $\mathcal{C}_3^{(s)}$, and $\Delta - 2$ roots of attached trees. By the induction hypothesis and Lemma A.1, the multiset of round-$s$ neighbor colors at $u$ and $u'$ is identical, hence $c_{s+1}(u) = c_{s+1}(u')$. The same argument applies to $v$ and $v'$. For $w, a, c$, each has two neighbors in $\mathcal{C}_1^{(s)} \cup \mathcal{C}_2^{(s)}$ with the same pair of round-$s$ colors (up to permutation), and the same number of attached tree roots, hence they share the same round-$(s + 1)$ color. The attached trees remain synchronized by Lemma A.1. This proves the invariant for round $s + 1$, and in particular $c_t(u) = c_t(u')$ and $c_t(v) = c_t(v')$.

Since every $\mathsf{E}_t$ endpoint decoder reads only the pair $(h_u, h_v)$ produced by $t$ WL-equivalent rounds, $e^+$ and $e^-$ are indistinguishable for $\mathsf{E}_t$.

**Step 3.** Consider the line graph $L(G_t)$. The node corresponding to $e^+ = (u, v)$ is denoted $\widehat{e^+}$ and that for $e^- = (u', v')$ is $\widehat{e^-}$. The neighbors of $\widehat{e^+}$ are the edge-nodes incident to $u$ and to $v$ (excluding $(u, v)$), which form two cliques in $L(G_t)$ of size $\Delta - 1$ each. Among these neighbors the two edge-nodes $(u, w)$ and $(v, w)$ are present. On the $e^-$ side, the neighbors of $\widehat{e^-}$ are the edge-nodes incident to $u'$ and to $v'$ (excluding $(u', v')$), with the special neighbors $(u', a)$ and $(v', c)$.

Run 1–WL on $L(G_t)$ with uniform initial colors. By degree symmetry, the first refinement partitions the neighbor sets of both $\widehat{e^+}$ and $\widehat{e^-}$ into two types: the $\Delta - 2$ "light" edges incident to a degree-1 leaf in $G_t$, and one "heavy" edge incident to the core neighbor on each side. Denote these heavy neighbors by $h_u = (u, w)$ and $h_v = (v, w)$ around $\widehat{e^+}$, and by $h_{u'} = (u', a)$ and $h_{v'} = (v', c)$ around $\widehat{e^-}$.

At the next refinement, $h_u$ and $h_v$ generally receive different colors, because their endpoint neighborhoods in $G_t$ differ: the $u$-side and $v$-side are distinguishable once the multiset of colors coming from their attached trees propagates one round through $L(G_t)$, while $h_{u'}$ and $h_{v'}$ remain synchronized by the diamond symmetry. Consequently the multiset of neighbor colors of $\widehat{e^+}$ differs from that of $\widehat{e^-}$ after a constant number of WL rounds on $L(G_t)$. Hence there exists $t_0 \in \{1, 2\}$ such that a model in $\mathsf{L}_{t_0}$ separates $\widehat{e^+}$ and $\widehat{e^-}$.

Combining the three steps proves Theorem 3.1. □

**Theorem 3.2.** *(Iteration-gap family) There exists a family $\{G_k\}_{k \geq 1}$ and edges $e_k, e_k' \in E(G_k)$ such that (i) a model in $\mathsf{L}_2$ separates $\hat{e}_k$ and $\widehat{e_k'}$ for all $k$; (ii) every model in $\mathsf{E}_k$ assigns the same score to $e_k$ and $e_k'$.*

*Proof of Theorem 3.2.* We instantiate the hypothesis classes as in Section 3.2: $\mathsf{E}_k$ are $k$-layer 1-WL–equivalent endpoint decoders on $G$, and $\mathsf{L}_t$ are $t$-layer 1-WL–equivalent models on $L(G)$ followed by a 1-layer classifier. In line with DuoLink's initialization, models in $\mathsf{L}_t$ may use *bounded-radius edge-node features* $z_{\hat{e}}$ (e.g., proximity indices such as Common Neighbors), which depend on a constant-radius neighborhood of $(u, v)$ in $G$, independent of $k$. Endpoint decoders in $\mathsf{E}_k$ operate on node embeddings only, as defined in Section 3.2.

**Construction of $G_k$.** Fix $k \geq 1$ and $\Delta \geq 3$. Build $G_k$ using the two cores from Theorem 3.1, padded with regular trees to depth $k - 1$:

*Triangle side.* Create $u, v, w$ and edges $(u, v), (v, w), (w, u)$. Attach $(\Delta - 2)$ disjoint leaf roots to each of $u, v, w$ and replace each leaf by the root of an identical $(\Delta - 1)$-ary tree of depth $k - 1$ (every internal node has total degree $\Delta$). Mark $e_k = (u, v)$.

*Diamond side.* Create $u', v', a, c$ and edges $(u', v'), (u', a), (a, c), (c, v')$. Attach $(\Delta - 2)$ disjoint leaf roots to each of $u', v', a, c$ and replace each leaf by the root of an identical $(\Delta - 1)$-ary tree of depth $k - 1$. Mark $e'_k = (u', v')$.

**Property (ii): indistinguishability for $\mathsf{E}_k$.** Run 1–WL on $G_k$ with uniform initial colors. Exactly as in the proof of Theorem 3.1, the synchronized padding (Lemma A.1) implies that after $k$ rounds

$$c_k(u) = c_k(u') \qquad \text{and} \qquad c_k(v) = c_k(v').$$

Hence any $k$-layer endpoint decoder in $\mathsf{E}_k$ receives identical pairs $(h_u, h_v)$ and $(h_{u'}, h_{v'})$ (up to injective recodings of $c_k$), and must assign the same score to $e_k$ and $e'_k$. This proves (ii).

**Property (i): separation for $\mathsf{L}_2$ with bounded-radius edge features.** Initialize each edge-node $\hat{e} = (u, v)$ in $L(G_k)$ with the *Common Neighbors* feature

$$\mathrm{CN}(u, v) \; = \; |\{\, x \in V : (u, x) \in E \text{ and } (v, x) \in E \,\}|,$$

optionally concatenated with other bounded-radius indices. This feature depends only on the 2-hop neighborhood of $(u, v)$ in $G_k$, hence its radius is constant in $k$.

By construction, $e_k = (u, v)$ has a common neighbor $w$, so $\mathrm{CN}(u, v) \geq 1$. In contrast, $e'_k = (u', v')$ has no common neighbor, so $\mathrm{CN}(u', v') = 0$. Therefore the two edge-nodes $\widehat{e_k}$ and $\widehat{e'_k}$ have distinct initial features $z_{\widehat{e_k}} \neq z_{\widehat{e'_k}}$.

A model in $\mathsf{L}_2$ (indeed, even $\mathsf{L}_1$) followed by a 1-layer classifier can separate these two points in feature space. Concretely, a single linear classifier on $z_{\hat{e}}$ implements the threshold rule $\mathbf{1}\{\mathrm{CN}(u, v) \geq 1\}$, which outputs different labels for $\widehat{e_k}$ and $\widehat{e'_k}$. Hence (i) holds.

For every $k \geq 1$ we have constructed $G_k$ and marked edges $e_k, e'_k$ such that (ii) indistinguishability persists for all models in $\mathsf{E}_k$, while (i) separation is achieved by a constant-depth line-graph model $\mathsf{L}_2$ using bounded-radius edge features. This proves Theorem 3.2. $\qquad\square$

Now, we prove Proposition 3.3. First, we need a key lemma.

**Lemma A.2** (Identity layer on $L(G)$)**.** *In $\mathsf{L}_1$, there exist message/update parameters so that the single MPNN layer on $L(G)$ outputs*

$$h_{\hat{e}}^{(1)} = z_{\hat{e}},$$

*i.e., it copies the initial edge-node features to the post-layer embedding.*

*Proof of Lemma A.2.* Use an MPNN with self-loops on $L(G)$ and injective aggregation (e.g., sum). Set neighbor messages to zero and the self-message to the identity on $z_{\hat{e}}$, and choose the update to return its first argument. Then the aggregated message at $\hat{e}$ equals $z_{\hat{e}}$ and the update outputs $h_{\hat{e}}^{(1)} = z_{\hat{e}}$. This is a standard parameter choice within 1-WL–equivalent MPNNs. $\qquad\square$

**Proposition 3.3** (*Realizing proximity-index rules on $L(G)$*) *Let $h_{struct}(u, v) \in \mathbb{R}^p$ be any fixed collection of proximity indices computed from a bounded-radius neighborhood of $(u, v)$, for example common neighbors, Adamic–Adar, Local Path up to length $K$, or truncated Katz. Initialize each edge-node $\hat{e}$ in $L(G)$ with $z_{\hat{e}} = [\, h_{struct}(u, v) \,\|\, h_{attr}(u, v) \,]$. For any Boolean threshold rule $f$ on these indices there exists a model in $\mathsf{L}_1$ with classifier $\rho$ that realizes $f(h_{struct}(u, v), h_{attr}(u, v))$.*

*Proof of Proposition 3.3.* Let $f : \mathbb{R}^p \to \{0, 1\}$ be a Boolean threshold rule, so there exist $w \in \mathbb{R}^p$ and $\tau \in \mathbb{R}$ with

$$f(x) = \mathbf{1}\{\langle w, x \rangle \geq \tau\}.$$

Recall $z_{\hat{e}} = [\, h_{\mathrm{struct}}(u, v) \,\|\, h_{\mathrm{attr}}(u, v) \,]$. Apply one layer of $\mathsf{L}_1$ as in Lemma A.2 to obtain $h_{\hat{e}}^{(1)} = z_{\hat{e}}$. Define the 1-layer classifier $\rho : \mathbb{R}^{p+q} \to [0, 1]$ by $\rho(h) = \mathbf{1}\{\langle \tilde{w}, h \rangle \geq \tau\}$, where $\tilde{w} = [\, w \,\|\, 0_q \,]$ puts zero weight on the attribute block. Then for every edge-node $\hat{e}$,

$$\rho\left(h_{\hat{e}}^{(1)}\right) = \mathbf{1}\{\langle w, h_{\mathrm{struct}}(u, v) \rangle \geq \tau\} = f(h_{\mathrm{struct}}(u, v), h_{\mathrm{attr}}(u, v)).$$

Thus a model in $\mathsf{L}_1$ realizes $f$ on $L(G)$.

The bounded-radius assumption guarantees that computing $h_{\mathrm{struct}}(u, v)$ and $h_{\mathrm{attr}}(u, v)$ depends only on a constant-radius neighborhood in $G$, independent of graph size. No further property is required. $\qquad\square$

## B    FURTHER EXPERIMENTAL DETAILS

### B.1    LINE-GRAPH ADJACENCY VIA INCIDENCE MATRIX

Let $G = (V, E)$ be an undirected graph with $|V| = n$ and $|E| = m$. Define its *incidence matrix* $B \in \{0, 1\}^{n \times m}$ by

$$B_{u,e} = \begin{cases} 1, & \text{if node } u \text{ is an endpoint of edge } e, \\ 0, & \text{otherwise.} \end{cases}$$

Then the original graph adjacency $A$ can be recovered (up to self-loops) as

$$A = B\,B^\top - \operatorname{diag}(d), \quad d_u = \sum_e B_{u,e} = \deg(u).$$

More importantly, the line-graph adjacency $A_L \in \{0, 1\}^{m \times m}$ is given by $A_L = B^\top B - 2I_m$

$$(A_L)_{e,e'} = \begin{cases} 1, & e \neq e' \text{ share exactly one endpoint in } G, \\ 0, & \text{otherwise.} \end{cases}$$

Since $(B^\top B)_{e,e} = 2$ for each edge $e$, subtracting $2I_m$ removes self-loops and yields the correct line-graph structure.

### B.2    DUOLINK VS. OTHER LINE-GRAPH APPROACHES

Table 5 summarizes the design differences between DuoLink and representative line-graph methods. Here, "Heuristics" denotes initializing edge-nodes on $L(G)$ with classical proximity indices and training over them end to end, "Attr. sim." refers to explicit attribute-similarity features (e.g., cosine of node embeddings), and "WL separation" indicates theoretical results showing expressivity or iteration-gap advantages for line-graph models over endpoint decoders on $G$.

Table 5: Conceptual contrast with representative line-graph approaches. A checkmark indicates explicit support.

| Method | Uses $L(G)$ | LP as node-cls on $L(G)$ | Heuristics | Attr. sim. | WL separation | End-to-end |
|---|---|---|---|---|---|---|
| LGNN (Cai et al., 2021) | ✓ | ✓ | | | | ✓ |
| LGCL (Zhang et al., 2023) | ✓ | ✓ | | | | ✓ |
| LineDi2vec (Xing & Makrehchi, 2024) | ✓ | | | | | |
| **DuoLink (ours)** | ✓ | ✓ | ✓ | ✓ | ✓ | ✓ |

**Notes.**    LGNN and LGCL both operate on the line graph and treat LP as node classification, training with supervised objectives, but neither incorporates classical heuristics or explicit attribute-similarity features as trainable inputs. LineDi2vec leverages the line graph for edge embeddings in an unsupervised manner rather than supervised node classification on $L(G)$. In contrast, DuoLink combines edge-node initialization with heuristics and attribute similarity, supports GNN and transformer backbones on $L(G)$, and is the only approach providing WL-based theoretical guarantees tailored to edge tasks.

### B.3    SCALABILITY AND BATCHING

**Time and Space Complexity.**    Naïvely, each node $u \in \mathcal{V}$ of degree $k$, induces a $k$-complete subgraph in $L(G)$ where the vertices are the adjacent to $u$. As $k$-complete graph has $\binom{\deg(u)}{2}$ edges, constructing $L(G)$ by examining each vertex's adjacency list takes

$$\sum_{u \in V} \binom{\deg(u)}{2} = O(\sum_u \deg(u)^2) = O(m\,d_{\max}),$$

where $d_{\max}$ is $G$'s maximum degree and $m = |E|$. Memory usage is dominated by storing $A_L$, which in sparse form requires $O(m\,d_{\max})$ entries.

### B.4  GRAPH HOMOPHILY AND HETEROPHILY

For completeness, we recall two standard homophily metrics in a labeled graph $\mathcal{G} = (\mathcal{V}, \mathbb{E}, \mathcal{C})$, where $\mathcal{C} : \mathcal{V} \to \{1, \ldots, N\}$ assigns each node to a class:

- **Node homophily ratio**:   $H_n(\mathcal{G}) = \frac{1}{|\mathcal{V}|} \sum_{v \in \mathcal{V}} \frac{\left|\{u \in \mathcal{N}(v) : \mathcal{C}(u) = \mathcal{C}(v)\}\right|}{\deg(v)}$.

  This measures, on average, the fraction of same-class neighbors per node.

- **Edge homophily ratio**:   $H_e(\mathcal{G}) = \frac{\left|\{(u,v) \in \mathbb{E} : \mathcal{C}(u) = \mathcal{C}(v)\}\right|}{|\mathbb{E}|}$,

  i.e. the proportion of edges that connect nodes of the same class.

Both $H_n$ and $H_e$ lie in $[0, 1]$. By convention, $H_n \geq 0.5$ (or $H_e \geq 0.5$) indicates a *homophilic* graph, while lower values denote *heterophily* (Zhu et al., 2020; Luan et al., 2022). Recent works propose alternative and more nuanced homophily measures to capture class imbalance, multi-factor similarity, and higher-order interactions (Jin et al., 2022; Luan et al., 2023).

### B.5  LINK-PREDICTION SETTINGS

Let $\mathcal{G} = (\mathcal{V}, \mathbb{E}, \mathcal{X})$ be an undirected, unweighted graph with node set $\mathcal{V} = \{v_1, \ldots, v_n\}$, edge set $\mathbb{E} \subset \mathcal{V} \times \mathcal{V}$, and an attribute matrix $\mathcal{X} \in \mathbb{R}^{n \times m}$ whose $i$th row $\mathcal{X}_i$ is the $m$-dimensional feature vector of node $v_i$. We partition both nodes and edges into observed (old) and unobserved (new) subsets, $\mathcal{V} = \mathcal{V}_o \cup \mathcal{V}_u$   and   $\mathbb{E} = \mathbb{E}_o \cup \mathbb{E}_u$.

During training, we see only $\mathcal{G}_o = (\mathcal{V}_o, \mathbb{E}_o)$, and our goal is to predict whether each candidate pair in $\mathcal{V} \times \mathcal{V}$ belongs to $\mathbb{E}_u$. Depending on which nodes are allowed at test time, link prediction falls into three categories (Menon & Elkan, 2011):

1. **Transductive** $(\mathcal{V}_o = \mathcal{V})$. All nodes are known at train time, and we predict missing edges among them.

2. **Inductive** $(\mathcal{V}_o \cap \mathcal{V}_u = \emptyset)$. We score edges between entirely unseen nodes using only their features.

3. **Semi-inductive**. Test edges may involve one or two nodes from $\mathcal{V}_u$.

In this paper, we restrict our focus to the commonly used transductive setting. Extensions to inductive and semi-inductive tasks are straightforward and discussed in Appendix B.7.

### B.6  DUOLINK AND TRANSDUCTIVE SETTING

Following established protocols in link prediction (Kipf et al., 2016; Pan et al., 2021), we randomly partition the set of positive edges $E$ into 85%, 5%, and 10% splits for training ($E_{\text{train}}^+$), validation ($E_{\text{valid}}^+$), and testing ($E_{\text{test}}^+$), respectively, except for OGB datasets, which use their predefined splits. For each group, we sample an equal number of negative edges, node pairs not present in $E$, to form $E_{\text{train}}^-$, $E_{\text{valid}}^-$, and $E_{\text{test}}^-$. We denote the union of positives and negatives in each split as $E_{\text{train}} = E_{\text{train}}^+ \cup E_{\text{train}}^-$, $E_{\text{valid}} = E_{\text{valid}}^+ \cup E_{\text{valid}}^-$, and $E_{\text{test}} = E_{\text{test}}^+ \cup E_{\text{test}}^-$.

To construct the input graphs for our experiments, we adopt the standard transductive setting. The **training line graph** is built from the graph $G_{\text{train}} = (V, E_{\text{train}})$, which contains only the training edges. We generate its line graph $L(G_{\text{train}})$, where each node represents a training edge and adjacency reflects shared endpoints in $G_{\text{train}}$. Node features are computed for all candidate edges in $E_{\text{train}}$ (both positive and negative), including proximity indices and attribute similarities.

For evaluation, we construct the **test line graph** from the full graph $G_{\text{test}} = (V, E_{\text{train}} \cup E_{\text{valid}} \cup E_{\text{test}})$. The corresponding line graph $L(G_{\text{test}})$ provides the evaluation context for all test candidates. Specifically, we classify candidate edges from $E_{\text{test}}^+$ and $E_{\text{test}}^-$ as nodes in $L(G_{\text{test}})$ using their respective features. This procedure ensures that all predictions are made within the full observed graph, while training is restricted strictly to the training set, thus preventing any information leakage.

### B.7 EXTENSIONS TO INDUCTIVE SETTINGS

Although our experiments focus on the transductive scenario, DuoLink extends naturally to inductive and semi-inductive link prediction:

- **New nodes.** When a previously unseen node $u'$ arrives with feature $X_{u'}$, we compute its incident edge-nodes $e' = (u', v)$ for $v \in V$. We then assemble $\mathcal{N}_1(e')$ in $L(G)$ via on-the-fly similarity heuristics $h_{\text{struct}}(u', v)$ and prune to $S$ neighbors.
- **New edges.** To score a candidate edge $e' = (u', v')$, we compute $\Delta_{u'} \cap \Delta_{v'}$ using current node features and local topology, embed $e'$ and its sampled neighbor-edges, and apply the same GNN layers.

This procedure requires only local recomputation of heuristics and sampling, without retraining or global graph access.

### B.8 PROXIMITY INDICES FOR DUOLINK

To enrich our link prediction framework with informative structural signals, we incorporated a comprehensive set of classical and higher-order heuristic indices. These features have demonstrated utility across a wide range of tasks, particularly in sparse or heterophilic settings where node attributes may be unreliable or absent. In our DuoLink framework, we utilized these indices $\{I_{(u,v)}\}$ as initial node embeddings for the nodes $\{\eta_{(u,v)}\}$ for the line graph $L(G)$.

The following proximity indices were computed for each candidate edge (node pair):

- Shortest Path Length
- Number of 2-paths and 3-paths
- Jaccard, Salton, and Sorensen indices
- 3-Jaccard, 3-Salton, and 3-Sorensen (higher-order extensions)
- Adamic-Adar index
- Hub Promoted Index (HPI) and Hub Depressed Index (HDI)
- Cosine Similarity, L1, and L2 distances (Attribute similarity)
- Pearson Correlation
- Jaccard similarity for binary vectors.

### B.9 T-SNE VISUALIZATIONS

The t-SNE visualizations on the test sets (Figures 4a to 4d) illustrate the representational differences between raw heuristic proximity indices (left), conventional node-pair embeddings from SAGE using Hadamard products of node embeddings (center), and DuoLink-SAGE embeddings obtained via the line graph (right). While raw proximity heuristics carry some discriminative signal, positives and negatives remain partially mixed and diffuse. The standard SAGE embeddings collapse this structure further, leading to highly intertwined and indistinct clusters, hindering accurate edge classification.

In contrast, DuoLink-SAGE produces embeddings that exhibit distinct, well-separated clusters for positive (real) and negative (fake) edges. This substantial improvement confirms that reformulating link prediction as node classification on the line graph allows GNNs to directly refine heuristic structural cues via edge-centric message passing. Consequently, DuoLink embeddings capture meaningful higher-order patterns missed by conventional decoders, directly explaining the strong empirical performance gains observed across the benchmarks.

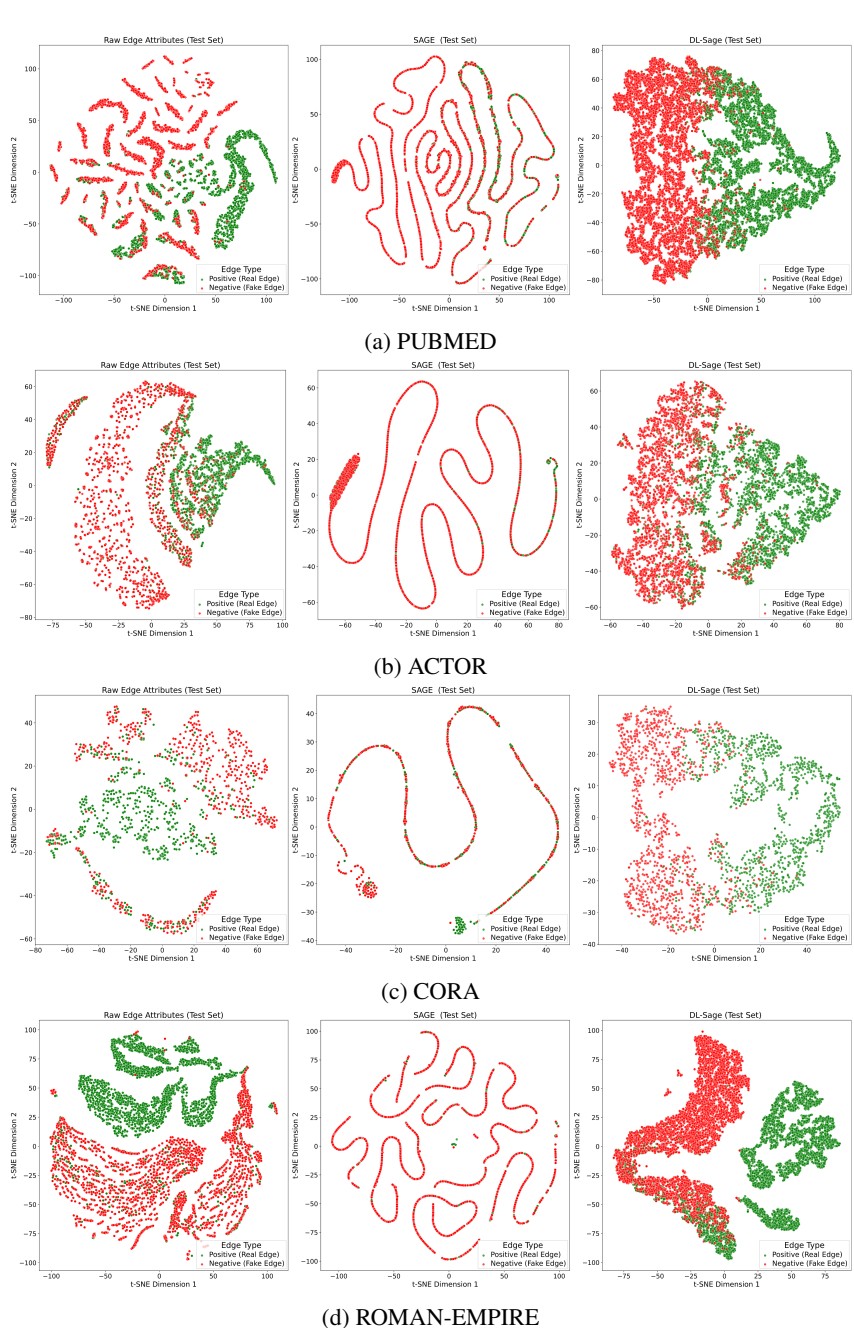

Figure 4: **t-SNE Visualizations** t-SNE visualizations of test-set edge representations for four datasets (Pubmed, Actor, Cora, Roman-Empire). Left: raw proximity heuristic features; middle: standard SAGE node-pair embeddings via Hadamard product; right: DuoLink-SAGE edge-centric embeddings on the line graph. Positive (real) and negative (fake) edges are colored separately. DuoLink-SAGE yields markedly better class separation, demonstrating how the line-graph reformulation and integrated structural heuristics enable clearer discrimination compared to both standalone heuristics and conventional decoded embeddings.

