# OpenReview forum: "DuoLink: A Dual Perspective on Link Prediction via Line Graphs"
_ICLR.cc/2026/Conference — ICLR 2026 Conference Withdrawn Submission_

### Official Review · Reviewer_yGmW · 2025-10-25

**Soundness:** 2
**Presentation:** 3
**Contribution:** 2
**Rating:** 2
**Confidence:** 5

**Summary:**

This paper studies the link prediction problem on both the homophilic and heterophilic graphs. The authors introduce, a line-graph formulation that casts link prediction as node classification task on the corresponding line graph. Line graph transformation translates the edges in the original graph to nodes in the line graph. Then if two edges in the original graph share a common end node, their corresponding nodes in the line graph form an edge. The edge-node is then initialized with proximity indices on the original graph and attribute similarity, which makes the proximity features trainable. Theorectical analysis shows that DuoLink has better expressiveness than 1-WL bounded GNN method for link prediction task. Experiments on both homophilic and heterophilic show the strength of DuoLink.

**Strengths:**

[S1] The paper is well motivated and demonstrated. It is easy to understand the benefits of transforming the original graph to line graph.

[S2] The performance advantage of DuoLink is significant compared to the baselines on the benchmarks in the experiment.

**Weaknesses:**

[W1] A few claims made by the paper is not rigor. For example,  the authors claim that decoder-based methods often perform worse than heuristics such as Common Neighbors or Adamic–Adar. This was true 3 years ago. However, a series of studies [1,2,3] have explored expressive ways to incorporate topological/structural features into the decoder-based models. Therefore, the claim about previous limitations should be made based on contemporary works, rather than the earlier work like GAE [4].

[W2] The novelty over LGNN[4] is limited. The major difference between DuoLink and LGNN is how the edge-node feature is initialized. In LGNN, the feature is initalized from original node's labeling trick. DuoLink is initialized from proximity indices, which is less expressive compared to node labelling.

[W3] The biggest concern I have is about the efficiency. In my understanding, during inference, all candidate links (both positive and negative) will be converted to edge-nodes in line graph. The possible number of negative links are quadractic to the number of nodes, which leads to computation overhead.

[W4] A minor but necessary point: DuoLink should also be tested on the four commonly-used OGBL datasets, including Collab, Citation2, PPA and ddi, for a comprehensive evaluation for the performance and efficiency benchmarking.


[1] Graph Neural Networks for Link Prediction with Subgraph Sketching. ICLR 23'.

[2] Pure Message Passing Can Estimate Common Neighbor for Link Prediction. Neurips 24'.

[3] Learning Scalable Structural Representations for Link Prediction with Bloom Signatures. WWW 24'.

[4] Line Graph Neural Networks for Link Prediction.

[4] Variational Graph Auto-Encoders.

**Questions:**

[Q1] Related to W3, is the line graph transform conditioned on each target link (like LGNN), or all the positive links and sampled negative links are converted to edge-node in a oneshot fashion for one batch of training edges?

---

### Official Review · Reviewer_iarZ · 2025-10-27

**Soundness:** 3
**Presentation:** 3
**Contribution:** 2
**Rating:** 4
**Confidence:** 3

**Summary:**

The paper proposes DuoLink, a framework that reformulates link prediction as node classification on the line graph L(G). Each edge becomes a node, initialized with proximity indices and optional attribute similarity, enabling message passing directly over edge neighborhoods. The approach aims to remove the encoder–decoder mismatch of traditional link prediction pipelines and theoretically demonstrates a 1-WL expressivity separation showing that shallow models on L(G) can distinguish edge motifs unreachable by bounded-depth decoders on G. Empirical results on both homophilic and heterophilic graphs show strong performance, often surpassing existing link-prediction GNNs and heuristics.

**Strengths:**

- Sound formulation: Casting link prediction as node classification on L(G) is elegant and theoretically well-motivated.
- Theoretical grounding: The WL-based analysis provides a solid justification for why line-graph message passing may offer higher expressivity for edge motifs.
- Empirical strength: The model performs well across both homophilic and heterophilic settings, improving over a broad range of baselines.
- Integration of heuristics: Turning classical proximity scores into trainable features is conceptually appealing and bridges traditional heuristics with GNN learning.
- Clear motivation and strong experiments: The work addresses a well-known gap between heuristics and GNNs for link prediction.

**Weaknesses:**

- Conceptual imprecision: In the introduction, the claim that “heterophilic graphs connect dissimilar nodes” is inaccurate.
In heterophilic graphs, connected nodes typically have different labels, but this does not necessarily mean they are feature-dissimilar,  a crucial conceptual distinction.
- Unclear statement: The claim that “similarity-based decoders implicitly assume homophily” is central but insufficiently explained.
A clearer, formal reasoning or empirical evidence supporting this would improve the paper’s clarity.
- Similarity to prior work: The proposed idea appears closely related to [1], which also addresses link prediction by recasting it as node classification. The authors should explicitly clarify how DuoLink differs from and advances beyond this approach.
- Experimental setup and scalability limitations: The datasets used (Cora, Citeseer, Pubmed) are not only relatively small for link prediction but also rely on simplistic and less meaningful edge splits, which often make the task artificially easy.
The more recent benchmark setting in [2] provides larger and more realistic splits, better reflecting real-world link prediction difficulty and generalization. Furthermore, while the line-graph reformulation is efficient on sparse graphs, its complexity can grow quadratically with graph density, since each node in the line graph corresponds to an edge in the original graph and edges in L(G) represent shared endpoints.
This can make the approach computationally prohibitive for denser datasets, and discussing such scalability limits would improve the paper’s completeness.
- Missing baselines: Recent strong models from [2,3] should be included for a fair and up-to-date evaluation.
- Expressivity context: The theoretical part would benefit from citing and discussing works like [4], which are directly relevant to the expressivity of GNNs.
- Limited theoretical depth: While the 1-WL separation result is formally correct, it remains somewhat narrow, it only demonstrates existence of cases where DuoLink on L(G) is more expressive than 1-WL on G, but does not characterize the overall expressivity or upper bounds. The analysis would benefit from a clearer connection to the broader theory of GNN expressivity (e.g., [4]) and from empirical validation showing that this theoretical advantage manifests in practice.
- The heterophilic datasets used in the experiments might not be the most suitable. As discussed in [5], these datasets present several issues that make them problematic for reliably evaluating GNN models. I would recommend using alternative datasets that do not suffer from such limitations.
- Minor repetition: The “Line Graph Transformation” section is repeated twice (Sec. 3), which disrupts the flow.

[1] Lachi, V. et al., A simple and expressive graph neural network based method for structural link representation, ICML 2024

[2] Li et al., Evaluating Graph Neural Networks for Link Prediction: Current Pitfalls and New Benchmarking, NeurIPS 2023

[3] Ferrini F. et al., GNNs Meet Sequence Models Along the Shortest-Path: an Expressive Method for Link Prediction, 2025

[4] Lachi, V. et al., Bridging Theory and Practice in Link Representation with Graph Neural Networks, NeurIPS 2025

[5] Platonov O. et al., A critical look at the evaluation of GNNs under heterophily: Are we really making progress?, ICLR 2023

**Questions:**

- Could you clarify the statement “similarity-based decoders implicitly assume homophily”? This seems conceptually important and deserves more elaboration.
- How does DuoLink differ from [1] in terms of novelty and modeling assumptions?
- Why not evaluate on larger and more consistent settings such as those proposed in [2]?
- How does your expressivity result relate to [4]?

---

### Official Review · Reviewer_VwDi · 2025-10-29

**Soundness:** 2
**Presentation:** 3
**Contribution:** 2
**Rating:** 2
**Confidence:** 5

**Summary:**

The paper proposes DuoLink, a link prediction framework that converts the problem into node classification on the line graph, where edges of the original graph are treated as nodes. Each edge-node is initialized using standard proximity-based heuristic scores and optionally attribute similarity. The authors argue that this formulation avoids the encoder–decoder separation common in GNN link prediction and allows message passing to focus directly on edge neighborhoods. They provide theoretical results suggesting that shallow models on the line graph can distinguish certain edge motifs that endpoint-based decoders cannot. Experimental results on a mix of homophilic and heterophilic graph benchmarks show improved performance over selected heuristic methods, standard GNN architectures, and recent link prediction models.

**Strengths:**

The paper is clearly written and easy to follow, with intuitive motivation and presentation of the proposed formulation. The method itself is conceptually simple and the problem addressed is relevant.

**Weaknesses:**

**Novelty**
The main limitation of the paper concerns the novelty of the contribution. The proposed idea of transforming the original graph into its line graph, formulating link prediction as node classification, and even providing theoretical expressivity results, has already been presented in [1]. The only substantial difference between DuoLink and [1] lies in the initialization of the nodes in the line graph: in [1], node features are obtained through an MLP applied to the concatenation of original node features of the edge (or they are structural features of nodes of the line graph when node features are not available in the original graph), while in this work the authors initialize edge-nodes with proximity indices. Although I recognize the usefulness of including these heuristic-based features and turn them into trainable features, changing the initialization alone does not seem to provide enough novelty to justify the paper’s contribution.

**Evaluation protocol**
First, the authors select negative links randomly, ensuring they are equal in number to the positive links. However, as shown in [2], random sampling produces negatives that are too easy to classify, resulting in an overly simplified and unrealistic evaluation of link prediction models. According to [2], the correct way to assess these models is to use HeaRT, which samples hard negative examples based on multiple heuristics. Considering these findings, I believe that the evaluation in this paper should either adopt HeaRT framework (which is already available for the homophilic dataset the authors used) or at least introduce a more realistic and challenging negative sampling procedure.

Second, there are important issues with the heterophilic datasets used for evaluation. As discussed in [3], classical heterophilic datasets such as Cornell, Texas, and Wisconsin present several serious limitations. Specifically, these graphs are extremely small (183–251 nodes and 280–499 edges), which leads to unstable and statistically insignificant results. In fact, prior works show that standard deviations on these datasets are very high. Moreover, these datasets have strong class imbalance; for example, the Texas dataset contains a class with only one node, which makes using that class for training or evaluation meaningless. On top of that, the authors evaluate performance using AUC, a metric known to be sensitive to class imbalance, making these results even less reliable. Therefore, evaluation on these three datasets is not recommended. More recent heterophilic datasets introduced in [3], such as Amazon-Ratings, Minesweeper, Tolokers, and Questions, are more suitable benchmarks for heterophilic link prediction and should have been considered (the authors only include Roman-Empire among them).

Finally, the evaluation on heterophilic graphs presents additional inconsistencies. Some of the baselines used (for example CFLP and LLP) are not actually designed specifically for heterophilic graphs. Moreover, it is unclear why only these “heterophily-specific” methods were tested on heterophilic datasets, while the other link prediction methods evaluated on homophilic datasets were not included here. Since the proposed method is also not explicitly designed for heterophilic graphs, it would be important to check whether non-specialized method can also perform well in this setting. This would provide a fairer and more comprehensive comparison, of course, assuming the evaluation is done on appropriate datasets as discussed above.

**Minor comment**
Since Theorem 3.1 is an existence theorem, it would be very helpful to include a concrete illustrative example of such a graph in a figure. Showing explicitly the edges $e^+$ and $e^-$ would make it much easier to understand the intuition behind the result. A graphical representation of the constructed graph and the relevant edges would convey the result far more clearly than the formal proof alone.


[1] Lachi, Veronica, et al. "A simple and expressive graph neural network based method for structural link representation." PROCEEDINGS OF MACHINE LEARNING RESEARCH 251 (2024): 187-201.

[2] Li, Juanhui, et al. "Evaluating graph neural networks for link prediction: Current pitfalls and new benchmarking." Advances in Neural Information Processing Systems 36 (2023): 3853-3866.

[3] Platonov, Oleg, et al. "A critical look at the evaluation of GNNs under heterophily: Are we really making progress?." The Eleventh International Conference on Learning Representations.

**Questions:**

1) Differences from [1]: What are the main differences between your method and [1]? In particular, from the theoretical perspective, how do your expressivity results differ from those already presented there? Are you able to position your contribution in a distinct way?

2) Feature combination: In cases where the original dataset provides node features, how are these used to build the features of the nodes in the line graph? More specifically, how are the original node features of incident nodes combined to form the one the the node in the line graph? and how is this combined with the proximity indices used for initialization?

3) Baselines on heterophilic datasets: Why do you test only a subset of link prediction methods on heterophilic graphs, while excluding the ones used for homophilic graphs? Since your model is not explicitly designed for heterophily, it would be important to see whether other generic methods can also perform well in this setting.

4) Clarity of Theorem 3.1 notation: In Theorem 3.1, the graph is denoted as $G_t$, but up to that point the paper never defines what a "graph with a subscript" actually represents. If the notation $G_t$ is meant to indicate an element of a family of graphs indexed by $t$, that should be stated explicitly.  However, in Theorem 3.1 the statement refers to a single graph, not to a family, so the use of a subscript is unclear and seems unnecessary. The confusion increases because the same subscript $t$ is also used to denote the number of GNN layers or WL iterations. It is not explained what the relationship is between this iteration depth and the graph $G_t$ itself; why should the graph depend on $t$? What property of the graph changes with $t$ that justifies calling it $G_t$?  This ambiguity remains in Theorem 3.2, where a family of graphs $\{G_k\}$ is introduced, again indexed by the number of layers. It would help to clearly separate the meaning of these indices (graph index vs.\ iteration depth) and to clarify whether the notation $G_t$ simply labels a specific constructed example or whether the graphs are actually parameterized by the model depth for some reason.


[1] Lachi, Veronica, et al. "A simple and expressive graph neural network based method for structural link representation." PROCEEDINGS OF MACHINE LEARNING RESEARCH 251 (2024): 187-201.

---

### Official Review · Reviewer_3DSj · 2025-10-31

**Soundness:** 3
**Presentation:** 3
**Contribution:** 2
**Rating:** 4
**Confidence:** 5

**Summary:**

This paper addresses the link prediction problem by proposing DuoLink, which is a link prediction method that combines a line representation of the graph with information from node similarity and classical link prediction heuristics. The line graph representation allows link prediction to be solved as a node classification problem using either a GNN or a graph transformer. Moreover, the line graph representation enables message passing over edges, which can be more convenient for capturing local structure, such as triangles. A theoretical analysis shows that there are edge patterns that are detectable by a shallow GNN on the line graph that cannot be detected by a fixed-depth GNN on the original graph. In the experiments, DuoLink is compared (using multiple GNN and graph transformer backbones) against state-of-the-art methods for link prediction using homophilic and heterophilic datasets, and the results show that it achieves consistent gains in accuracy.

**Strengths:**

- The paper is well-written and easy to follow

- The proposed solution is simple and achieves good results

- The proposed approach is evaluated using homophilic and heterophilic datasets

**Weaknesses:**

- The technical contributions are limited: The key idea of DuoLink is the use of the line graph representation. This idea has been proposed in the past, as stated in the paper. Is the novelty combining the line representation with the proximity features described in Section 2.2? Proximity features have also been applied by recent link prediction methods, but maybe not using the line graph representation. I just can’t see how the combination of two ideas from the literature is sufficient for publication in a venue such as ICLR.

- The line graph representation is not well-justified: As stated in the paper, the line graph representation adds computational time and lacks theoretical support. The paper shows that the line graph representation can do things that a fixed-width GNN cannot do,  but one can simply increase the width. The converse is also not discussed: can the line representation do everything that the original representation can do?  For instance, there are non-isomorphic graphs that are indistinguishable by the line representations. From a more practical point of view, the line graph representation seems to require knowledge of the candidate edges during the graph construction phase (the nodes must exist to be classified). In practice, there are too many candidate edges. Moreover, information from candidate edges can leak test information to the training.

- Important experimental results are missing: The paper discusses running times in the text, but a clear training time comparison should be provided (including the graph construction phase). Moreover, the line graph representation should be evaluated against a vanilla GNN and the baselines without the proximity features (to separate its effect from the rest). Experiments with larger datasets (e.g., from OGBL) are also needed. Finally, several papers criticize the link prediction problem setup used in this paper (see paper below, for example):
https://arxiv.org/pdf/2405.14985

**Questions:**

1) Is the novelty of the paper combining the line graph representation with proximity features?

2) Is the line graph representation as powerful as the original graph representation?

3) How to make sure that information from the test set (candidate edges) do not leak to the training phase?

4) How are training times of DuoLink compared with the baselines?

5) How much of the improvements in the results come from the line graph representation vs. the structural features?

---

### Note · Authors · 2025-11-12

**Comment:**

We thank the reviewers for their valuable time and feedback. We plan to revise the paper by incorporating their suggestions and will submit it to another venue.

Best wishes,

The Authors

**Withdrawal Confirmation:**

I have read and agree with the venue's withdrawal policy on behalf of myself and my co-authors.